# The Potential Mechanism of *Alpiniae oxyphyllae* Fructus Against Hyperuricemia: An Integration of Network Pharmacology, Molecular Docking, Molecular Dynamics Simulation, and In Vitro Experiments

**DOI:** 10.3390/nu17010071

**Published:** 2024-12-28

**Authors:** Shuanggou Zhang, Yuanfei Yang, Ruohan Zhang, Jian Gao, Mengyun Wu, Jing Wang, Jun Sheng, Peiyuan Sun

**Affiliations:** 1Key Laboratory of Pu-er Tea Science, Ministry of Education, Yunnan Agricultural University, Kunming 650201, China; zhang15012106135@163.com (S.Z.); 15215003760@163.com (R.Z.); 18214585880@163.com (J.W.); shengj@ynau.edu.cn (J.S.); 2College of Food Science and Technology, Yunnan Agricultural University, Kunming 650201, China; 3College of Science, Yunnan Agricultural University, Kunming 650201, China; YYF0209@126.com (Y.Y.); gj1902916072@163.com (J.G.); 15208717558@163.com (M.W.)

**Keywords:** *Alpiniae oxyphyllae* fructus, hyperuricemia, network pharmacology, molecular docking, molecular dynamics simulation, in vitro experiments

## Abstract

**Background**: *Alpiniae oxyphyllae* Fructus (AOF) is a medicinal and edible resource that holds potential to ameliorate hyperuricemia (HUA), yet its mechanism of action warrants further investigation. **Methods**: We performed network pharmacology, molecular docking, molecular dynamics simulation, and in vitro experiments to investigate the potential action and mechanism of AOF against HUA. **Results**: The results indicate that 48 potential anti-HUA targets for 4 components derived from AOF were excavated and predicted through public databases. Gene Ontology (GO) enrichment analysis indicated that there are 190 entries related to biological process, 24 entries related to cellular component, 42 entries related to molecular function, and 44 entries related to Kyoto Encyclopedia of Genes and Genomes (KEGG) signaling pathways. The results of molecular docking showed that the main active ingredients of AOF may have potential therapeutic effects on immune system disorders and inflammation caused by HUA by binding to targets including peroxisome-proliferator-activated receptor gamma (PPARG), estrogen receptor 1 (ESR1), prostaglandin G/H synthase 2 (PTGS2), and 3-hydroxy-3-methylglutaryl-coenzyme A reductase (HMGCR). Subsequently, we further determined the stability of the complex between the core active ingredient and the core target proteins by molecular dynamics simulation. The results of cell experiments demonstrated that stigmasterol as the core active ingredient derived from AOF significantly upregulated the expression levels of ESR1 and PPARG (*p* < 0.001) to exert an anti-HUA effect. **Conclusions**: In summary, we have systematically elucidated that the mechanism of main active ingredients derived from AOF mainly exert their pharmacological effects by acting on multiple targets in this study. Our studies will provide a scientific basis for the precise development and utilization of AOF.

## 1. Introduction

Hyperuricemia (HUA) is a metabolic disease caused by increased purine metabolism, elevated uric acid (UA) levels, or impaired excretion in the human body, resulting in excessive accumulation of UA in the blood [1,2]. A substantial body of evidence indicates that the disease not only causes gout but is also closely linked to chronic arthritis, joint deformities, uric acid kidney stones, hypertension, diabetes, cardiovascular disease, and various other chronic conditions [3,4,5]. The prevalence of HUA in China has been increasing year by year [6]. At present, the drugs mainly used for the clinical treatment of HUA include allopurinol, benzbromarone, probenecid, febuxostat, and colchicine [7,8,9].

Despite their potential benefits, these conventional medicines can cause significant adverse reactions during the treatment process, which limits their clinical application [10,11]. Therefore, there is an urgent need to develop more new therapeutic strategies for HUA. There are rich and diverse plant resources within the unique natural environment of China, providing our country with countless natural foods and medicinal resources [12]. With the growing awareness of health at the national level, these food and medicinal resources have become an important direction for the development of the big health industry, favored by researchers both domestically and internationally, and have become a research hotspot in recent years. Consequently, developing natural products derived from these food and medicinal resources that can ameliorate HUA holds significant potential for research and development.

*Alpiniae oxyphyllae* Fructus (AOF) is the dried and mature fruit of the high-altitude plant *Alpinia oxyphyllae*, valued both as a medicinal and edible resource [13]. It is mainly grown in some southern regions of China, such as Yunnan, Hainan, Guangxi, Guangdong, and Fujian. Studies have shown that some active ingredients derived from AOF exhibit anti-oxidant, anti-inflammatory, anti-cancer, neuroprotective, anti-diarrheal, and anti-gastric damage effects [14,15,16,17]. In recent years, some researchers have found that AOF also has a certain positive effect on HUA [18]. Although its specific functional mechanism is rarely reported, the combination of bioinformatics and network pharmacology provides a practical approach for exploring their mechanisms of action [19].

Network pharmacology is a research method based on systems biology theory, one that utilizes network mining data and integrates a large amount of information to explore drug targets and molecular mechanisms [20,21]. Since 2017, it has become one of the mainstream methods in the field of traditional Chinese medicine research [22].

Molecular docking is a computer simulation method used to study the interactions between molecules, predict small molecule ligands and suitable target binding sites, and predict the binding mechanism and affinity between the two [23,24]. It is of great significance for further understanding the interactions between compounds and targets, as well as drug discovery and development. Molecular dynamics simulation is a technology that focuses on the motions resolved over time between receptors and small molecules. It is utilized to further determine and assess the binding of ligands to receptor proteins, thereby providing a more precise understanding of dynamics and interaction [25,26]. This study used network pharmacology to explore the active ingredients of AOF; predict its targets for HUA; and verify the prediction results using molecular docking technology, molecular dynamics simulation techniques, and in vitro cell experiments. The research results provide a scientific basis for the development and utilization of AOF products, and they also provide more research ideas for the prevention and treatment of HUA.

## 2. Materials and Methods

### 2.1. Software and Database

The potential targets of AOF were obtained from the Traditional Chinese Medicine System Pharmacology (TCMSP) database (https://www.tcmsp-e.com/,accessed on 6 April 2024), Swiss Target Prediction database (http://www.swisstargetprediction.ch/, accessed on 8 April 2024), and UniProt database (https://www.uniprot.org/, accessed on 8 April 2024). The GeneCards database (https://www.genecards.org/, accessed on 6 April 2024) was utilized to predict the gene targets related to HUA. The targets acquired above all require the removal of duplicate values. Venny 2.1.0 (https://bioinfogp.cnb.csic.es/tools/venny/, accessed on 10 May 2024) was used to screen for common targets of AOF and HUA. Finally, we imported the obtained common targets into the STRING database (https://string-db.org/, accessed on 10 May 2024), DAVID database (https://david.ncifcrf.gov/, accessed on 10 May 2024), Cytoscape Software (Version 3.10.0), and WeChat Analytics Platform (http://www.bioinformatics.com.cn/, accessed on 16 May 2024) for visual analysis. The specific process is shown in Figure 1.

### 2.2. Materials and Instruments

Human kidney proximal tubule cell line (HK-2) was provided by the Kunming Cell Bank of the Chinese Academy of Sciences, and stigmasterol (purity assayed by HPLC: 95.0%) was purchased from Yuan ye Biotechnology (Shanghai, China). Dulbecco’s modified Eagle’s medium/Friendship Mixture F-12 (F12/DMEM) culture medium and fetal bovine serum (FBS) were purchased from Thermo Fisher Scientific (Pittsburgh, PA, USA), while peroxisome-proliferator-activated receptor gamma (PPARG) and estrogen receptor 1 (ESR1) rabbit monoclonal antibodies were purchased from Abimat Medical Technology Co., Ltd. (Shanghai, China). The electrophoresis instrument (DYY-6D) was purchased from the Beijing Liuyi Instrument Factory, and the exposure machine (FluorChem) was purchased from Protein Simple (San Jose, CA, USA).

### 2.3. Determination of Active Ingredients in AOF

All chemical components of AOF were retrieved from the TCMSP Database. Then, based on oral bioavailability (OB) ≥ 30% and drug like (DL) ≥ 0.18, the main potential active ingredients of AOF were screened. The TCMSP Database and Swiss Target Prediction Database were used to obtain the potential targets of the drug active ingredients. After screening and removing duplicate targets, they were entered into the Uniprot Database one by one to standardize the target protein gene names and finally obtain the target information of the active ingredients of AOF.

### 2.4. Searching for the Active Ingredients of AOF

Gene targets related to hyperuricemia were retrieved using the keyword “hyperuricemia” in the GeneCards database; we performed a screening and removed duplicate targets, and the Venny 2.1.0 plotting tool was used for Venny analysis to establish the intersection target of AOF and HUA.

### 2.5. Construction of the Protein–Protein Interaction (PPI) Network

The potential targets of AOF for treating HUA were imported into the STRING Data Analysis Platform, and then the species as Homo sapiens and the minimum medium confidence were set to 0.40. We exported the analysis results in TSV format, and then we used Cytoscape 3.10.0 software to draw the PPI network. We visualized the analysis using the “Network Analyzer” section of the software to obtain the core target interaction map.

### 2.6. Construction of the “Drug Ingredient Target Disease Network Diagram”

The screened active ingredients and disease target genes of AOF were input into Cytoscape 3.10.0 software to construct a drug ingredient target disease network diagram. Then, we used the “Network Analyzer” section of the software to analyze network topology parameters such as betweenness centrality, centrality, and degree.

### 2.7. GO and KEGG Enrichment Analysis of Targets

In order to further investigate the role of drug disease intersection targets in gene function, parameters were set to OFFICIAL_ENE_SYMBOL, Homo saplens, and Gene List in the DAVID database for GO analysis and KEGG pathway enrichment analysis. The analysis results were downloaded, and the top ten pathways were selected based on *p* < 0.05 as the standard and visualized using a bioinformatics analysis platform.

### 2.8. Molecular Docking

The key active ingredients of AOF and the core target were selected from the intersection targets for molecular docking experiments. We completed the pre-treatment and molecular docking of key active ingredients and core targets in AutoDock Tools v 1.56 software, calculated the docking binding energy, and evaluated the docking results. We then visualized and analyzed the docking results using PyMol Viewer 1.5 software, adjusted the results at appropriate angles, and analyzed the interactions between ligands and receptors. Finally, we saved the images.

### 2.9. Molecular Dynamic Simulation

The optimal docking model generated by molecular docking was utilized to perform a molecular dynamics simulation via the Gromacs-2020.6 package. The AMBER99SB force field and the simple point charge (SPC) water model were selected to assign the receptor proteins. Subsequently, we set a cubic box to place the receptor proteins, and the surface charges for the receptors were subsequently neutralized by adding counterions (Na^+^ or Cl^−^). In order to relax all atoms of the receptor proteins, the steepest descent and conjugate gradients method were set at 5000 steps to perform energy minimization. Then, the minimized system was gradually heated by a normal volume and temperature (NVT) ensemble at 300 K for 1 ns, followed by a normal pressure and temperature (NPT) ensemble at 300 K for 1 ns. Finally, the molecular dynamics (MD) system was run for 30 ns to analyze the dynamic binding mode of the complexes.

### 2.10. Cell Culture

HK-2 cells were cultured in F12/DMEM medium containing 10% fetal bovine serum and 1% penicillin–streptomycin, and they were placed in a 37 °C incubator containing 5% CO_2_. The medium was changed every two days until the cell fusion degree reached 80–90% for further experiments.

### 2.11. Cytotoxicity Experiment

The toxic effect of stigmasterol on the growth of HK-2 cells was determined using the methyl thiazolyl tetrazolium (MTT) assay. We transferred 2 × 10^4^ cells per well into a 96-well plate. After 24 h, we replaced the culture medium with serum-free medium containing different concentrations of stigmasterol (0, 1, 2, 4, 8, 16 μM). After incubation for 24 h, we added 20 μL MTT solution to each well and incubated them in the dark for 4 h to remove the supernatant. We then added 200 μL dimethyl sulfoxide (DMSO) to each well and shook them for 10 min. We measured their absorbance levels at 492 nm using an enzyme-linked immunosorbent assay (ELISA) reader.

### 2.12. Western Blotting Asssay

Western blot (WB) was used to detect the expression levels of key target proteins PPARG and ESR1 in HK-2 cells. In short, an equal amount of total protein was separated on 10–12% sodium dodecyl sulfate polyacrylamide gel (SDS-PAGE) and transferred to the PVDF membrane. After membrane transfer, the membrane was sealed with 5% skim milk at room temperature for 1 h, then detected overnight with primary antibodies (PPARG, ESR1) at 4 °C, washed 3 times with TBST buffer, and incubated with appropriate HRP-conjugated secondary antibodies at room temperature for 1 h. Finally, imaging and photography were performed in an exposure machine.

## 3. Results

### 3.1. Screening of Active Ingredients and Targets of AOF

A total of 41 active ingredients of AOF were detected on the TCMSP platform. Four main active ingredients were screened based on OB ≥ 30% and DL ≥ 0.18, namely, stigmasterol, sitosterol, daucosterol, and sitosterol palmitate, as shown in Table 1. Using TCMSP and Swiss Target Prediction Databases to search for compound targets, 276 targets were obtained after removing duplicates, as shown in Figure 2.

### 3.2. AOF Improved Potential Target Prediction of Hyperuricemia

Using the GeneCards Database to search for HUA-related targets, 928 targets were obtained after deduplication screening. Using the Venny 2.1.0 drawing platform, the intersection of the active ingredient targets of AOF and the HUA-related target genes was obtained, resulting in 46 target genes, as shown in Figure 3.

### 3.3. Construction of Intersection Target Network Between AOF and HUA

Using Cytoscape 3.10.0 software, the 46 pieces of intersecting target information of the active ingredient targets of AOF and HUA disease targets were visualized to obtain the target active ingredient disease network, as shown in Figure 4. This network had 254 edges and 54 nodes, and the number of lines connected by one node is called a degree. The larger the node, the higher its degree value, and the stronger the correlation. Among them, the active ingredients of AOF (stigmasterol, sitosterol, daucosterol, and sitosterol palmitate) are depicted in green, while the targets are depicted in blue. Using the built-in “Network Analyzer” function to analyze network topology parameters, the average topology coefficient was 0.6139, the average node betweenness was 0.0203, the average centrality was 0.4917, and the average degree value was 4.7037. This indicates that the various active ingredients of AOF mainly exert their therapeutic effects by acting on multiple targets, fully reflecting the characteristics of AOF’s intervention in HUA with multiple components and targets.

### 3.4. Analysis of PPI Network Construction

We imported the information of 46 intersecting targets between drugs and diseases into the STRING Database (reliability greater than 0.4) for PPI protein interaction analysis (Figure 5A), and we plotted them using Cytoscape 3.10.0 software. The color intensity of the target varied with the number of targets it interacted with, as shown in Figure 5B. The network consisted of 46 nodes and 392 edges, with each node representing a single protein target and edges representing the interaction relationship between proteins. The node size was proportional to the degree value, and the saturation of the color map also increased accordingly. In the PPI network, the average connectivity was 11.40, the average betweenness centrality was 0.0278, and the average compactness was 0.4653. The top four in terms of connectivity were PPARG, PTGS2, ESR1, and HMGCR. The specific information of key targets is shown in Table 2.

### 3.5. GO Enrichment Analysis of Potential HUA Targets in AOF

GO analysis was performed on the intersection targets of AOF and HUA using the DAVID Database, resulting in a total of 190 biological processes (BP), including positive regulation of cell proliferation, exogenous stimulus response, negative regulation of cell apoptosis, negative regulation of pri-miRNA transcription by RNA polymerase II, and negative regulation of gene expression. There were 24 cellular components (CC), including receptor complexes, macromolecular complexes, the cytoplasmic membrane, extracellular vesicles, and the endoplasmic reticulum. In Molecular Function (MF), a total of 44 entries were enriched, including protein binding, norepinephrine binding, protein homodimerization activity, enzyme binding, RNA polymerase II transcription factor activity, ligand-activated sequence-specific DNA binding, and oxidoreductase activity. Using *p* < 0.05 as the screening criterion, the top ten items were selected for visual analysis, and the results are shown in Figure 6.

### 3.6. KEGG Pathway Enrichment Analysis of the Potential Anti-HUA Targets of AOF

KEGG enrichment mainly studies the role of target proteins in signaling pathways. Through analysis, a total of 44 pathways related to 46 intersecting targets were screened, and cancer and other disease-related pathways with low *p* values (*p* < 0.05) and high rankings were selected. The top 10 KEGG rich set analysis bubble plots were plotted, as shown in Figure 7. The signaling pathways mainly involved included the cancer pathway, serotonin synaptic pathway, cancer transcription disorder pathway, proteoglycan pathway in cancer, chemical carcinogenesis receptor activation pathway, EGFR tyrosine kinase inhibitor resistance pathway, PI3K Akt signal pathway, ovarian steroidogenesis pathway, VEGF signal pathway, lipid and atherosclerosis, etc.

### 3.7. Analysis of Molecular Docking

Molecular docking is a method of studying molecular interactions, primarily focusing on the interactions between receptors and ligands, as well as calculating their binding modes and affinities [27]. In order to verify the interaction between key targets and compounds, we selected four main active ingredients of AOF as ligands, which were stigmasterol, sitosterol, daucosterol, and sitosteryl palmitate. The top four key proteins with degree values, PPARG (PDB ID: 6MS7), ESR1 (PDB ID: 1XPC), PTGS2 (PDB ID: 5F1A), and HMGCR (PDB ID: 1DQ9), were used as receptors for docking. The docking results are shown in Table 3. Generally speaking, if the binding energy is less than 0 kcal/mol, it is considered to be able to dock in a natural state. If it is less than −4.25 kcal/mol, it indicates a certain binding force. If it is less than −5.0 kcal/mol, it indicates good binding between the two, and if it is less than −7.0 kcal/mol, it indicates strong binding ability between the two [28,29,30]. In this study, hydrogen bonds were formed between various active ingredients and protein targets, with the strongest binding between stigmasterol and PPARG, with binding energies of −9.38 kcal/mol. The binding affinity between carotenoids and PTGS2 was the strongest, with binding energies of −5.13 kcal/mol. As shown in Figure 8, hydrophobic interactions can be formed between stigmasterol and Leu536, Leu354, Trp383, Ala350, Lys531, and Cys530 of ESR1; hydrogen bonds can be formed between stigmasterol and Glu343; and residues Leu469, His323, Tyr327, Met364, Ile326, Ala292, Leu333, and Ile341 formed hydrophobic interactions with the compound. These results indicate that the active ingredients of AOF can bind well with the key protein targets of HUA, further verifying the accuracy of network pharmacology screening.

### 3.8. Verification of the Binding Models by Molecular Dynamic Simulation

To further determine the stability of the interactions between stigmasterol and the core target proteins, we performed molecular dynamic simulations for 30 ns on the two optimal binding models, which were the complexes of stigmasterol–ESR1 and stigmasterol–PPARG. As depicted in Figure 9A,B, the root mean square deviation (RMSD) for the complex of stigmasterol–ESR1 or stigmasterol–PPARG reached stability for 20–30 ns. Overall, the details of the RMSD plots and the stability of the ligand–receptor complexes was close to that of free ESR1 or PPARG with fluctuation of less than 0.1 nm. As shown in Figure 9C,D, the radius of gyration (Rg) plots for the complexes of stigmasterol–ESR1 and stigmasterol–PPARG remained stable after 20 ns throughout the molecular dynamic simulation, suggesting the two complexes became compact forms. The above results indicated that our predicted models are credible.

### 3.9. The Effect of Stigmasterol on UA-Induced Inhibition of HK-2 Cells

Stigmasterol is a kind of phytosterol, one that mainly comes from herbs, soybeans, etc., and it is the main active ingredient in *Alpinia oxyphylla*. As a tetracyclic alkyl compound, it contains double bonds and multiple hydroxyl groups (Figure 10A), so it has anti-inflammatory, anti-diabetes, blood-lipid- and glucose-reducing, and other biological activities. In the UA-induced HK-2 cell experiment, it was found that low concentrations of stigmasterol had no toxicity to HK-2 cells, and at 2 μM and 4 μM, stigmasterol showed a certain proliferative effect on HK-2 cells (Figure 10B). Subsequently, we treated cells with 80 μg/L of UA, and then we determined the effect of different concentrations of stigmasterol on UA-induced cell activity. The results showed that various doses of stigmasterol, particularly at 2 μM and 4 μM, significantly increased cell viability in a concentration-dependent manner compared with the model group (Figure 10C). Therefore, 2 μM and 4 μM were selected as the optimal concentrations for subsequent experiments.

### 3.10. The Effect of Stigmasterol on the Expression of Key Target Proteins in HK-2 Cells

Based on the results of network pharmacology, molecular docking, and molecular dynamics simulation, AOF was found to hold good binding ability with the key target proteins such as ESR1 and PPARG. Therefore, it is speculated that AOF may ameliorate HUA through these two targets. To verify this hypothesis, we established a UA-induced HK-2 cell model. As is well known, the kidneys are the main excretory organ of UA, and HUA is mainly caused by the obstruction of UA excretion. In order to ensure the successful establishment of the UA-induced HUA model, we measured the UA content in the supernatant of HK-2 cells. The results showed (Figure 11A) that compared with the control group, the UA content in the model group was significantly reduced, indicating the successful establishment of the HUA model, and the addition of stigmasterol significantly increased the excretion of UA. After the successful establishment of the HUA model, we performed a Western blot assay to detect the effect of stigmasterol on the expression of ESR1 and PPARG. As shown in Figure 11B–D, after HUA induction, the expression of ESR1 and PPARG significantly decreased, while the addition of 2 μM and 4 μM stigmasterol significantly increased the expression of the two key target proteins, and the effect of 4 μM was significantly better than that of 2 μM. These results are completely consistent with the results of molecular docking and molecular dynamics simulation, indicating that AOF mainly ameliorates HUA by regulating the expression levels of the two key proteins, ESR1 and PPARG.

## 4. Discussion

HUA is caused by the increase in UA concentration in the blood, which not only increases the risk of gout but also is related to hypertension, diabetes, cardiovascular disease, metabolic syndrome, and other diseases [31,32,33]. In recent years, the incidence of HUA and various clinical complications has rapidly increased, having become one of the important public health issues in the international community [34]. Studies have shown that AOF has a certain preventive and therapeutic effect on HUA, but the specific mechanism is still unclear, which hinders the further development and utilization of AOF and its related products.

This study searched the TCMSP Database using the keyword “*Alpiniae oxyphyllae*” and obtained a total of 41 active ingredients. Four main active ingredients were screened based on OB and DL for target prediction, resulting in a total of 276 targets. It can be inferred that these four compounds may be the main active ingredients of AOF in intervening hyperuricemia by constructing a target active ingredient disease network diagram. Through further analysis, stigmasterol has 45 potential targets, sitosterol has 54 potential targets, daucosterol has 106 potential targets, and sitosteryl palmitate has 71 potential targets. Among them, stigmasterol, sitosterol, and sitosteryl palmitate are plant sterols that cannot be synthesized by the human body and can only be obtained from plants [33]. At present, stigmasterol has been proven to be an immunomodulator with great therapeutic potential [32]. It is widely studied for its anti-diabetes, antioxidant, anti-cancer, anti-inflammatory, antiviral, antiparasitic, anti-osteoarthritis, antibacterial, neuroprotective, and immunomodulatory properties, and it plays a role through different mechanisms [34,35,36]. Daucosterol is a kind of saponin phytosterol, one that has pharmacological properties such as potential antioxidant, anti-diabetes, hypolipidemic, anti-inflammatory, and immune regulation effects [34]. Given that the four main active ingredients of AOF have multiple pharmacological effects, they can play an important synergistic role in improving high UA level.

After analyzing the PPI network, key protein targets such as PPARG, ESR1, PTGS2, and HMGCR were identified, suggesting that AOF may ameliorate HUA through these targets. Research has shown that in addition to reducing UA level, antioxidant stress and anti-inflammatory effects may also contribute to the treatment of HUA [32]. PPARG is a nuclear receptor that binds to peroxisome proliferators, and by regulating its expression, it can effectively alleviate oxidative stress and improve HUA. PTGS2 is a bicyclic oxygenase and peroxidase in the prostaglandin biosynthesis pathway, mainly involved in regulating inflammatory responses and cytokine production involved in immune responses. Monosodium urate (MSU) crystals can increase the synthesis of prostaglandin E2 (PGE2) by increasing the expression of PTGS2, thereby causing or exacerbating inflammation such as gout [35]. Hence, HUA exceeding normal levels can lead to pathological changes such as inflammation, oxidative stress, and cell apoptosis. It can be inferred that the active ingredients of AOF may exert pharmacological effects such as antioxidant, anti-inflammatory, and immune regulation effects by acting on the key targets mentioned above.

Based on the results of GO and KEGG enrichment analysis, it can be inferred that the gene functions of the active ingredients in AOF mainly reflect basic functions such as cell growth, proliferation, metabolism, apoptosis, and signal transduction, which are conducive to exerting normal cellular functions, reducing oxidative stress and inflammation, and thus improving immunity. In addition, stigmasterol activates pro-apoptotic proteins, triggers the PI3K/Akt and VEGF signaling pathways [37,38,39,40], and is involved in various types of cancer. Daucosterol plays an important anti-cancer role in many signal pathways, such as inhibition of the PI3K/Akt pathway, cell cycle progression, and atherosclerosis [28]. In addition, the results of molecular docking showed that the main active ingredients of AOF can effectively bind to key protein targets of HUA, which indirectly confirms the reliability of previous network pharmacology analysis. Among them, stigmasterol has the strongest binding ability to PPARG and ESR1. The two key targets are closely related to oxidative stress and inflammation, which may indicate the importance of PPARG and ESR1 in the anti-HUA efficacy of AOF. In addition, the results of molecular dynamics simulation also indicate that stigmasterol can tightly bind with PPARG and ESR1.

Therefore, we further validated that stigmasterol mainly improves HUA through PPARG and ESR1 at the cellular level. The results showed that low concentrations of stigmasterol had no toxicity to HK-2 cells and significantly reversed UA-induced cell damage. In addition, stigmasterol significantly promoted UA excretion by increasing the expression levels of ESR1 and PPARG in a dose-dependent manner. These results are consistent with network pharmacology and molecular docking simulation predictions, confirming the feasibility of AOF in ameliorating HUA.

## 5. Conclusions

In summary, the four main active ingredients in AOF may alleviate HUA by regulating 46 key targets, including PPARG, PTGS2, ESR1, and HMGCR, involving 24 cellular components, 44 molecular functions, 190 cellular processes, and 44 signaling pathways. This study constructed an interaction network architecture between AOF and HUA through network pharmacology and explored the potential therapeutic targets and molecular mechanisms of AOF on HUA by combining molecular docking simulations and in vitro experiments. Our results will provide a certain application basis and scientific basis for the prevention and treatment of HUA.

## Figures and Tables

**Figure 1 nutrients-17-00071-f001:**
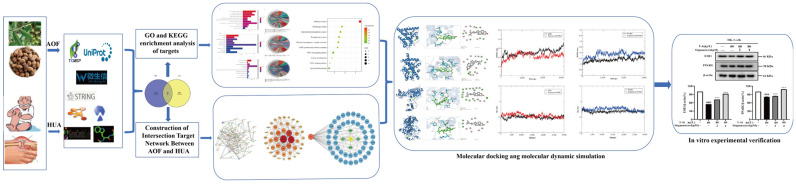
Network pharmacology flowchart of AOF for improving HUA. ### *p* < 0.001 vs. the control; *** *p* < 0.001 vs. the UA.

**Figure 2 nutrients-17-00071-f002:**
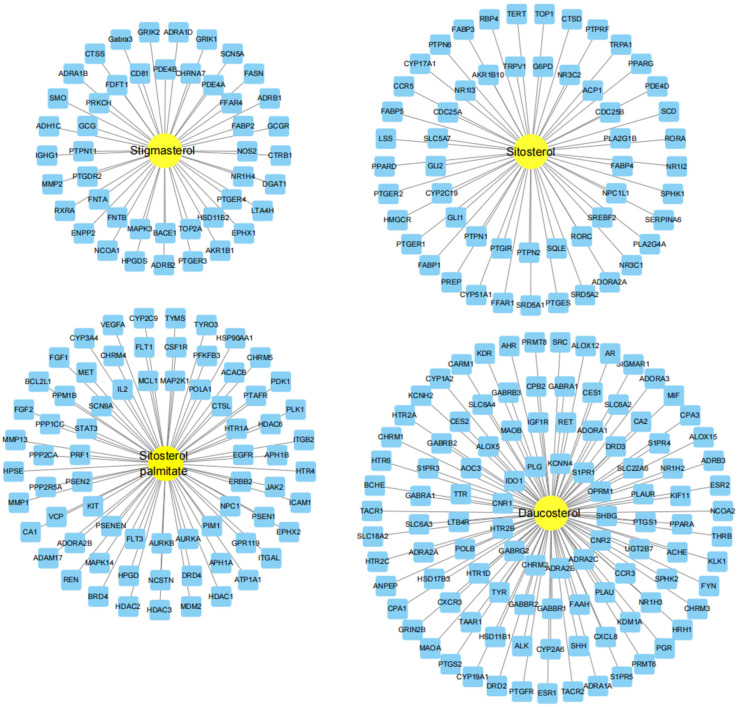
Target network of the active ingredients derived from AOF.

**Figure 3 nutrients-17-00071-f003:**
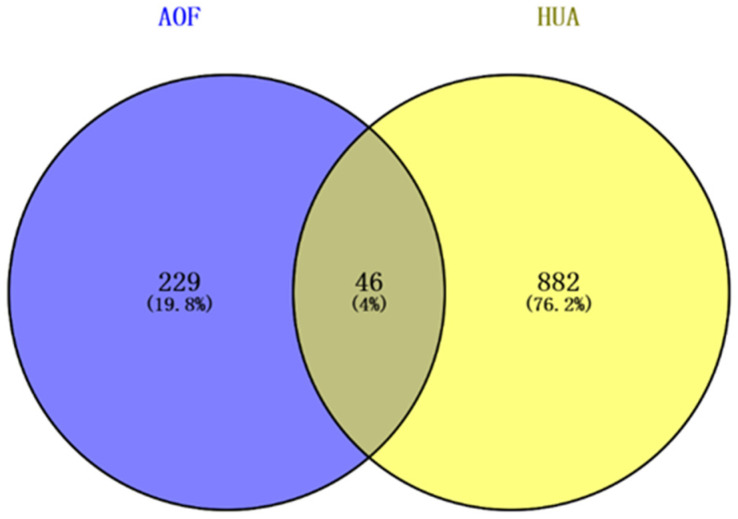
The overlap between the targets associated with active components of AOF and the targets associated with HUA depicted by a Venn diagram.

**Figure 4 nutrients-17-00071-f004:**
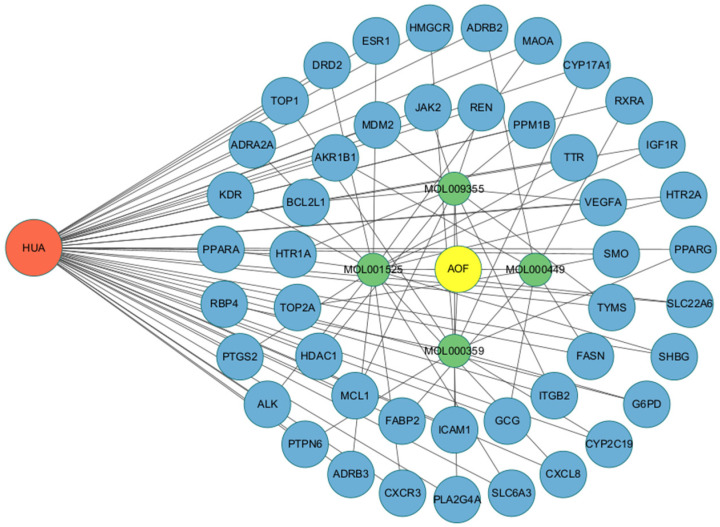
Network diagram of target-active ingredient disease for the anti-HUA efficacy of AOF.

**Figure 5 nutrients-17-00071-f005:**
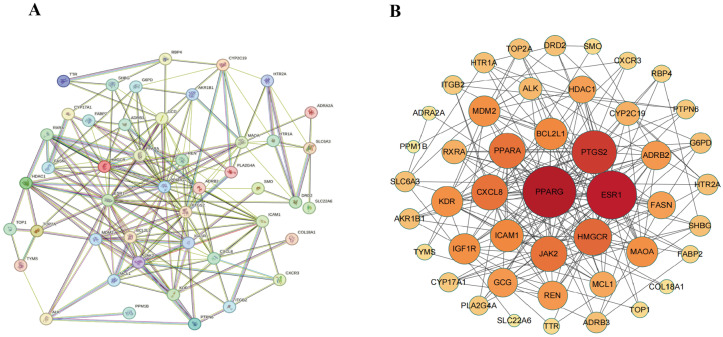
The PPI network of targets between HUA and AOF was constructed. (**A**) The PPI network diagram was derived from the STRING database. (**B**) Cytoscape software was utilized to identify the core targets within the protein interaction network. Nodes represent proteins, and edges indicate the connections between proteins. The size and color of the nodes correspond to the significance of each protein within the overall network.

**Figure 6 nutrients-17-00071-f006:**
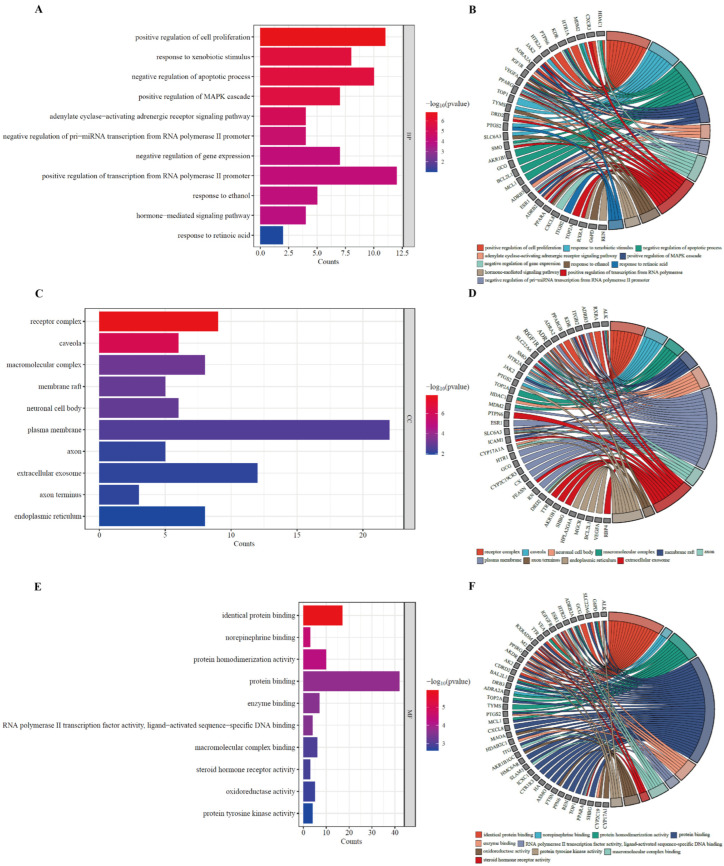
GO enrichment analysis (**A**,**C**,**E**) and the core targets against HUA (**B**,**D**,**F**).

**Figure 7 nutrients-17-00071-f007:**
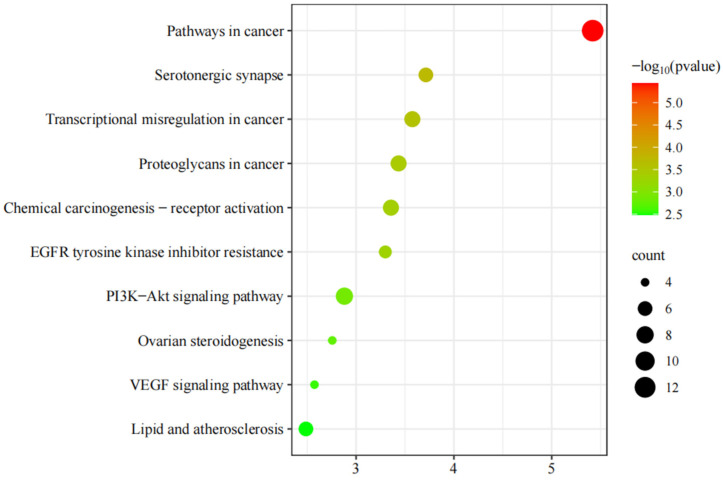
KEGG pathway enrichment analysis of potential targets for AOF intervening with HUA.

**Figure 8 nutrients-17-00071-f008:**
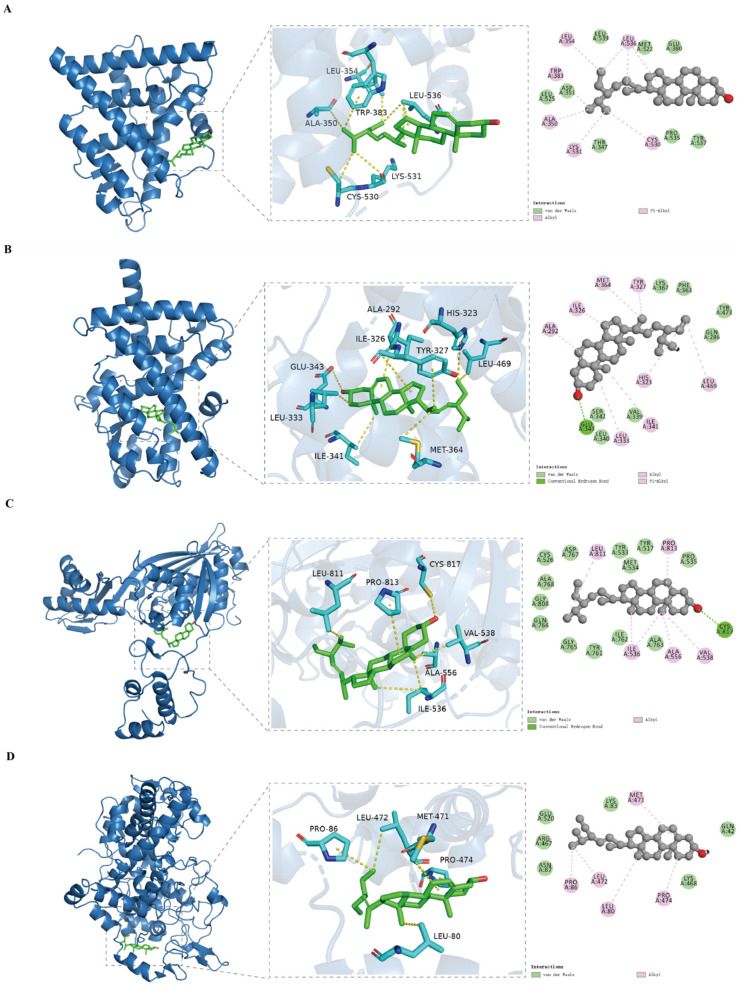
The docking conformation between the main active ingredients of AOF and some key targets of HUA. Figures (**A**–**D**) display the molecular docking outcomes of stigmasterol with the following proteins: ESR1 (**A**), PPARG (**B**), HMGCR (**C**), and PTGS2 (**D**).

**Figure 9 nutrients-17-00071-f009:**
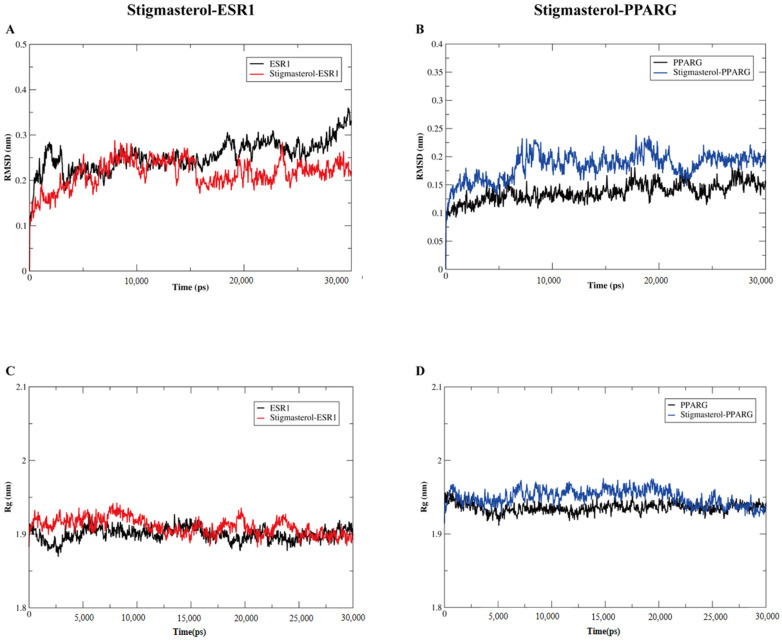
Molecular dynamics simulation for stigmasterol bound to ESR1 or PPARG by using Gromacs 2020.6. (**A**) The RMSD plots for free ESR1 and the complex of stigmasterol–ESR1. (**B**) The RMSD plots for free PPARG and the complex of stigmasterol–PPARG. (**C**) The Rg plots for free ESR1 and the complex of stigmasterol–ESR1. (**D**) The Rg plots for free PPARG and the complex of stigmasterol–PPARG.

**Figure 10 nutrients-17-00071-f010:**
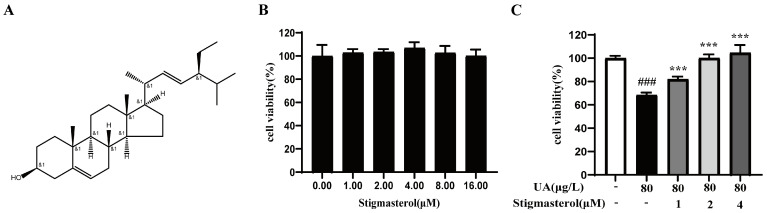
The cytotoxicity and anti-HUA effect of stigmasterol on HK-2 cells. (**A**) The structural formula of stigmasterol. (**B**) Effects of different concentrations of stigmasterol on the cell viability of HK-2 cells. (**C**) Different concentrations of stigmasterol attenuated the inhibitory effect induced by UA on HK-2 cells. ### *p* < 0.001 vs. the control; *** *p* < 0.001 vs. the UA.

**Figure 11 nutrients-17-00071-f011:**
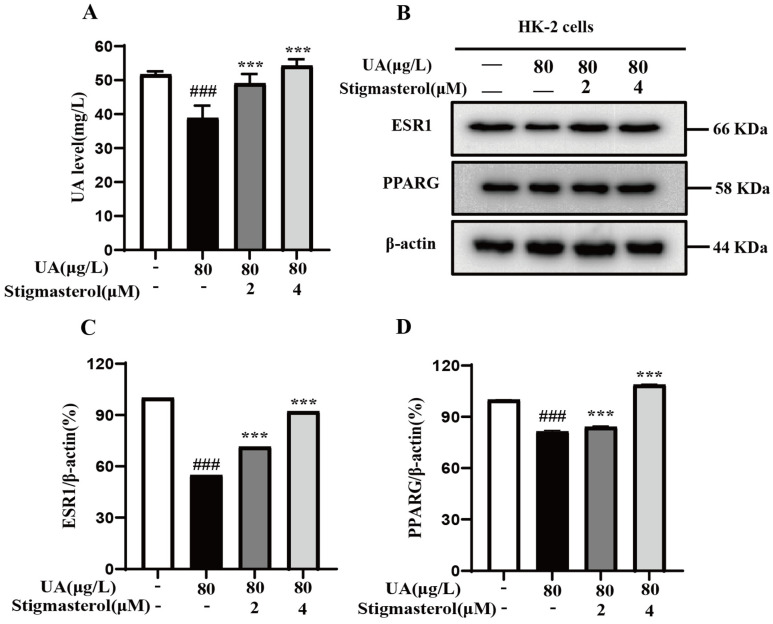
Stigmasterol exhibited anti-HUA effects by promoting UA excretion via regulating ESR1 and PPARG. (**A**) Stigmasterol increased UA excretion in HK-2 cells. (**B**) Effects of different concentrations of stigmasterol on the expression of key target proteins were detected by Western blotting. (**C**) The protein expression levels of ESR1 were quantified. (**D**) The protein expression levels of PPARG were quantified. All values are expressed as the average of three independent experiments (mean ± SEM). ### *p* < 0.001 vs. the control; *** *p* < 0.001 vs. the UA.

**Table 1 nutrients-17-00071-t001:** The active ingredients derived from AOF.

Mol ID	Ingredient Name	OB/%	DL
MOL000449	Stigmasterol	43.83	0.76
MOL000359	Sitosterol	36.91	0.75
MOL001525	Daucosterol	36.91	0.75
MOL000355	Sitosterol palmitate	30.91	0.44

**Table 2 nutrients-17-00071-t002:** Basic information of key targets.

No.	Key Core Targets	Neighborhood Connectivity	Closeness Centrality	Betweenness Centrality
1	PPARG	11.08	0.6618	0.1813
2	ESR1	11.70	0.6618	0.1834
3	PTGS2	12.55	0.6081	0.0757
4	HMGCR	11.75	0.5625	0.0512

**Table 3 nutrients-17-00071-t003:** Prediction of binding energy between the main active ingredients of AOF and key protein targets related to HUA.

Chemical Compound	CAS	Average Binding Energy (kcal/mol)			
		PPARG	ESR1	PTGS2	HMGCR
Stigmasterol	83-48-7	−9.38	−8.73	−6.43	−7.49
Sitosterol	83-46-5	−8.94	−8.50	−5.93	−5.01
Daucosterol	474-58-8	−4.01	−3.05	−5.13	−2.52
Sitosteryl palmitate	2308-85-2	−2.59	−2.56	−3.06	−0.89

## Data Availability

Data will be made available on request.

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
