# Peer review of "The Potential Mechanism of Alpiniae oxyphyllae Fructus Against Hyperuricemia: An Integration of Network Pharmacology, Molecular Docking, Molecular Dynamics Simulation, and In Vitro Experiments"

_nutrients, 2024, doi:10.3390/nu17010071_

Round 1
Reviewer 1 Report
Comments and Suggestions for Authors
Integrating network pharmacology, molecular docking, molecular dynamics simulations, and in vitro experiments is a robust approach that ensures multi-faceted validation of results. The study links AOF’s effects to critical targets such as PPARG and ESR1, which are well-known regulators of inflammation and metabolic pathways, strengthening the biological plausibility of the findings.
Areas for Improvement:
- While in vitro experiments demonstrated significant effects, more quantitative data on dose-dependent responses or IC50 values could enhance the depth of analysis.
- The exact molecular mechanisms by which stigmasterol modulates ESR1 and PPARG require further elucidation, particularly in vivo.
Best wishes,
Author Response
Comments 1: While in vitro experiments demonstrated significant effects, more quantitative data on dose.dependent responses or lC50 values could enhance the depth of analysis.
Response 1: Thank you very much for your constructive suggestion. We have conducted additional experiments based on your suggestion, including the result of Figure 10C and Figure 11A, hoping to further demonstrate the molecular mechanism.
Comments 2: The exact molecular mechanisms by which stigmasterol modulates ESR1 and PPARG require further elucidation, particularly in vivo.
Response 2: Thank you very much for your constructive suggestion. We fully agree with your suggestion. In our revised manuscript, we have determined in vitro that stigmasterol exhibits anti-HUA efficacy by promoting UA excretion via the two targets, ESR1 and PPARG. According to the reviewer’s comment, we also plan to use in vivo experiments to validation and even to explore more molecular mechanisms in the future. We appreciate the reviewer for the guidance.
Reviewer 2 Report
Comments and Suggestions for Authors
The idea of this paper is good. Also, it is very correctly written and organized.
However, there are some minor changes to be made before acceptance:
- in vitro should be in Italic face;
- Introduction: it is not clearly stated if the authors would like to refer to anti-HUA medication approved only in China, or all over the world (I would suggest the second one); still, there are some drugs (also approved in China) that are missing, like febuxostat and colchicine, that should be included;
- page 2 - line 58: Alpiniae oxyphyllae fructus (AOF) should be in Italic face and "fructus" not capitalized; also, AOF (a fruit basically) can NOT be a substance; it only can contain one or more substances;
- page 2 - line 59: I would maybe change "produced" with "grown";
- page 2 - line 70: Enter before "Molecular docking" because a new paragraph, about a different subject, is beginning;
- page 3 : 2.2. Material and Instruments: authors should use Past tense instead of Present tense;
- page 4: charges (Na+ and Cl-) should be in superscript;
-Table 1: title: delete "were excavated".
Author Response
Comments 1: in vitro should be in ltalic face.
Response 1: Thank you sincerely for your suggestion. We have made modifications to the original text based on your suggestions.
Comments 2: Introduction: it is not clearly stated if the authors would like to refer to anti-HUA medication approved only in China, or all over the world (l would suggest the second one); still, there are some drugs (also approved in China) that are missing, like febuxostat and colchicine, that should be included.
Response 2: Thank you very much for your comments and suggestions. We fully agree with your suggestion and have made modifications in the corresponding part.
Comments 3: page 2 - line 58: Alpiniae oxyphyllae fructus (AOF) should be in ltalic face and "fructus" notcapitalized; also, AOF (a fruit basically) can NOT be a substance: it only can contain one or moresubstances.
Response 3: Thank you very much for your comments despite your busy schedule. We have made corresponding modifications in the corresponding part based on your suggestion.
Comments 4: page 2 -line 59: I would maybe change "produced" with "grown".
Response 4: Thank you very much for your suggestion. We have replaced the term ‘produced’ with “grown” in the corresponding parts of the revised manuscript.
Comments 5: line 70: Enter before "Molecular docking" because a new paragraph, about a different subject, is beginning.
Response 5: Thank you sincerely for your suggestion. We have made revisions to the original manuscript based on your suggestions.
Comments 6: 2.2. Material and Instruments: authors should use Past tense instead of Present tense.
Response 6: Thank you very much for your constructive suggestion. We have changed the tense of the Material and Instruments section to the past tense.
Comments 7: page 4: charges (Na+ and Cl-) should be in superscript.
Response 7: Thank you very much for your comments despite your busy schedule. We have changed the charges of Na+ and Cl- to superscript in the original manuscript.
Comments 8: Table 1: title: delete "were excavated".
Response 8: Thank you very much for your suggestion. We have removed "were excavated" based on your suggestion.